Visual assessment of dynamic knee joint alignment in patients with patellofemoral pain: an agreement study

Hansen Rudi jth299@alumni.ku.dk 1
Lundgaard-Nielsen Mathilde 1
Henriksen Marius 2
1 Department of Physical and Occupational Therapy, Copenhagen University Hospital Bispebjerg and Frederiksberg , Copenhagen , Denmark
2 The Parker Institute, Copenhagen University Hospital Bispebjerg and Frederiksberg , Copenhagen , Denmark
Mendez-Rebolledo Guillermo
Electronic publication date: 2021 Oct 19
Publication date: 2021
Volume: 9
Electronic Location ID: e12203
Received 2021 Jun 17; Accepted 2021 Sep 2
Copyright: ©2021 Hansen et al.
Copyright year: 2021
Copyright holder: Hansen et al.
License: This is an open access article distributed under the terms of the Creative Commons Attribution License, which permits unrestricted use, distribution, reproduction and adaptation in any medium and for any purpose provided that it is properly attributed. For attribution, the original author(s), title, publication source (PeerJ) and either DOI or URL of the article must be cited.
License URL: https://creativecommons.org/licenses/by/4.0/

Keywords: Agreement, Knee Alignment, Patellofemoral Pain, PFP, Visual Assessment, Reliability, Knee Kinematics

Funding: The authors received no funding for this work.

==============================
Background

Assessment of knee kinematics plays an important role in the clinical examination of patients with patellofemoral pain (PFP). There is evidence that visual assessments are reliable in healthy subjects, but there is a lack of evidence in injured populations. The purpose of this study was to examine the intra- and interrater agreement in the visual assessment of dynamic knee joint alignment in patients with PFP.

Methods

The study was a cross-sectional agreement study. Sixty participants (42 females) were included. We assessed the intra- and interrater agreement of two functional tests: The single leg squat (SLS) and the forward lunge (FL). One investigator scored the movement according to preset criteria while video recording the movement for retest. Moreover, the performance was scored by another investigator using the video recording. Agreement was assessed using weighted kappa statistics.

Results

The intrarater agreement ranged from moderate to good (Kappa 0.58 (FL) to 0.70 (SLS)) whereas the interrater agreement ranged from fair to moderate (Kappa 0.22 (SLS) to 0.50 (FL)).

Conclusion

The agreement within raters was better than between raters, which suggests that assessments should preferably be performed by the same tester in research and in a clinical setting, e.g., to evaluate any treatment effect. We promote FL as a reliable clinical tool for evaluating dynamic knee alignment, since it shows equally good intra- and interrater agreement, and it is an inexpensive and easy method to use.

Background

Malalignment of the lower extremity have been linked to musculoskeletal problems, including patellofemoral pain (PFP) (Myer et al., 2015; Willson, Binder-Macleod & Davis, 2008; Powers, 2003; Gwynne & Curran, 2018). During movements such as squatting, individuals with PFP have demonstrated greater knee abduction excursion than controls (Nakagawa et al., 2012; Willson & Davis, 2008), and improvements in frontal and transverse plane pelvis and hip control have been linked to a reduction in pain (Mascal, Landel & Powers, 2003). Therefore, physiotherapists use visual assessment of dynamic alignment in their clinical decision-making process when considering prescription of exercises (Stephen et al., 2020).

Dynamic alignment can be measured by clinical observation or biomechanical motion-analysis technology. Development of three-dimensional biomechanical analyses has made it possible to quantify and evaluate knee kinematics during functional tasks in a valid and reliable manner (Mok, Petushek & Krosshaug, 2016). However, this method is generally costly and too time consuming in the clinical setting. Two-dimensional measures, such as the frontal plane projection angle and visual assessments or ratings of lower extremity motion, have been suggested as more cost effective and acceptable alternatives to three-dimensional motion capture (Willson & Davis, 2008; Harris-Hayes et al., 2014).

In visual assessments of frontal plane knee motion during a single leg squat (SLS) and a forward lunge (FL), the reliability is reported to range from moderate to excellent in nonsymptomatic subjects (Harris-Hayes et al., 2014; Weeks, Carty & Horan, 2012; Stensrud et al., 2011; Ageberg et al., 2010; Jones et al., 2014; Tate et al., 2015). Measurement properties of a test instrument are likely to depend on the population studied, and there is a lack of evidence in injured populations (Whatman, Toomey & Emery, 2019). In PFP patients reliability measures have been evaluated using a 2-D video capture procedure that quantifies frontal plane knee alignment during single limb squats (Gwynne & Curran, 2014). The results of this study suggest that this method is reliable (ICC = 0.86) in PFP patients. Measures of visual assessment of lower extremity kinematics in PFP patients without the use of video analysis have been limited to a single study evaluating a lateral step-down task (Piva et al., 2006). This study included the evaluation of several aspects of movement quality (arm strategy, trunk movement, pelvis plane, knee position and balance) and scored each item according to a scale designed for the study to assess an overall movement quality. While this multimodal approach to evaluate movement quality may be more comprehensive, it is difficult to compare the results to other more commonly used measures of alignment and to extrapolate the findings to a clinical setting.

Because therapists use visual ratings to make clinical decisions, the reliability of these ratings needs to be considered. Our intention was to evaluate whether a simple visual rating of knee movement during two commonly used tests of dynamic alignment (the SLS and FL test) can be used reliably among PFP patients. The rating method used in this study resembles a clinical setting where the therapists do not have access to the equipment or time required for complex biomechanical analysis. The SLS and the FL were chosen because they are commonly used in clinical practice and have been reported in many previous studies investigating visual rating of lower extremity function (Whatman, Hume & Hing, 2013). The tests are less demanding than the commonly used jump tests; they also more closely resemble activities of daily life, such as stair ascending/descending, which may be more appropriate in a population of both sports-active and sedentary individuals.

The aims of this study were (a) to determine the intrarater agreement of a subjective visual assessment by an experienced sports physiotherapist in evaluating dynamic knee control during a SLS and a FL in a group of PFP patients, and (b) to determine the interrater agreement of the subjective assessment of dynamic knee control between two experienced sports physiotherapists.

Materials & Methods

Study design

The study was a cross-sectional agreement study. The reporting of the study follows the Guidelines for Reporting Reliability and Agreement Studies (GRRAS) (Kottner et al., 2011).

Participants

Participants were a subset of an RCT study aiming to compare the effectiveness of therapeutic hip and knee exercise for patients with PFP and to identify candidate patient characteristics that predict differential responses to the two exercise programs (clinicaltrials.gov identifier: NCT03069547). The study was pre-planned in the parent trial protocol in order to assess measurement properties of the dynamic knee joint alignment measures, that will be assessed as a potential patient characteristic associated with treatment response. The assessments were performed at baseline in the parent trial, i.e., before randomization. As part of the baseline information gathered in the parent trial the participants answered the Anterior Knee Pain Score (a specific disability score for PFP patients ranging from 0 to 100 points, where higher scores indicate less disability (Kujala et al., 1993)), and self-reported pain during the past 4 weeks assessed on a 0 (‘no pain’) to 10 (‘worst imaginable pain’) Numeric Rating Scale. One participant failed to complete the Anterior Knee Pain Score and 3 participants failed to answer one of the 13 questions. The reported mean is calculated for 59 participants and missing data handled by imputing the mean of the participants for that particular item as recommended by Hott et al. (2021) and Chavance (2004). Participants was recruited from the Institute of Sports Medicine Copenhagen (ISMC), Bispebjerg-Frederiksberg Hospital, Denmark. ISMC is a medical unit mainly for patients with injuries in the musculoskeletal system caused by participation in sports activities, and thus most participants are sports active. We aimed at including 60 participants in this sub-study, which gives 80% power to detect a kappa-coefficient of at least 0.5 that is statistically significantly different from 0 and corresponds to a moderate agreement.

Raters

Raters were RH (male) and MLN (female) who are both sports physiotherapists with 18 and 15 years of experience, respectively, in treating and assessing patients with musculoskeletal problems. Both raters use visual assessments as part of their daily clinical practice but have not used it for research purposes.

General procedures

Video was recorded using a tablet (Apple Ipad Air 2, frame rate: 30 frames per second) from an anterior view of participants performing the SLS and FL in the gym at the Department of Physical and Occupational Therapy at Bispebjerg and Frederiksberg Hospital. An investigator (RH) instructed the participants to perform the selected movement as described below. After the instruction had been understood, the participant performed the selected movement without rehearsal. If the participant lost his/her balance during the test, a new trial was performed. No discussion of the testing procedures or the classification criteria occurred during the testing. The investigator filmed the participant and simultaneously scored the movement as observed clinically according to the criteria set below. The tablet was set up directly in front of participants, perpendicular to the frontal plane at a height of 100 cm and a distance of 3.5 m. The video captured the whole person. At least 1 week later, the investigator did another scoring based on the recorded video and another investigator (MLN) repeated the scoring independently. Three playbacks of the recorded video in real time were allowed for the intra- and interrater assessment.

Knee alignment during single leg squat

The SLS test has been described in several studies and the present procedure was a replication of comparable agreement studies (Harris-Hayes et al., 2014; Ageberg et al., 2010; Nae et al., 2017). From a position of single leg standing (painful knee), individuals performed a partial squat on one leg (hip and knee flexion) with the trunk maintained in an upright position, the contralateral hip in neutral and contralateral knee flexed. Individuals were instructed to perform the squat until they reached maximum ankle dorsiflexion without lifting their heels and then return to upright standing (Fig. 1A). The SLS was performed at participant-selected speed.

Knee alignment during forward lunge

The FL test was performed according to comparable agreement studies (Nae et al., 2017; Alkjaer et al., 2002). From a position of bilateral stance, individuals performed a forward step (painful knee), and continued the motion by flexing the front and back knee simultaneously (forward lunge).

Figure 1 Screenshots of video recordings in the assessment of a single leg squat (A) and a forward lunge (B).

Individuals were instructed to continue the lunge until reaching maximum dorsiflexion of the stance leg without lifting their heel and to push-off to upright position (Fig. 1B). The FL was performed at participant-selected speed. For both tests, dynamic valgus alignment was defined as an excessive medial movement of the knee as evidenced by an apparent increased frontal plane knee angle during the selected movement. Varus alignment was defined as an excessive lateral movement.

Scoring

The rater determined if a dynamic worsening of valgus or varus was present and scored the movement using the following categories:

−4 = severe valgus

−3 = moderate valgus

−2 = mild valgus

−1 = doubtful valgus

0 = no evidence of neither valgus nor varus

1 = doubtful varus

2 = mild varus

3 = moderate varus

4 = severe varus

We defined ‘no evidence of neither valgus nor varus’ as a neutral knee alignment, i.e., knee flexion aligned with the 2nd toe. We considered ‘doubtful’ a just merely detectable deviation from neutral alignment, while ‘mild’, ‘moderate’ and ‘severe’ was considered a definite deviation from neutral. The raters ranked the deviations ‘mild’, ‘moderate’ or ‘severe’ based on the experience of the raters in assessing PFP patients. We defined ‘mild’ as a slight deviation that might not be clinically relevant, ‘moderate’ as a modest and clinically relevant deviation from neutral, and ‘severe’ as a clinically relevant and severe collapse of the knee.

Similar ordinal and nominal scales have previously been used in the assessment of intra- and interrater agreement (Weeks, Carty & Horan, 2012; Chmielewski et al., 2007; Junge et al., 2012; Trulsson, Garwicz & Ageberg, 2010; Whatman, Hing & Hume, 2012). Visual assessments of dynamic knee joint alignment have been validated against a ‘gold standard’, i.e., three-dimensional motion analysis, and the ability of visual assessments to identify ‘true’ malalignment is considered acceptable (Ageberg et al., 2010; Whatman, Hume & Hing, 2013).

Single leg squat and forward lunge classifications

An a priori categorical classification was made where the scores −4 to −2 were categorized as ‘Definite valgus present’, the scores −1, 0 and 1 were categorized as ‘No definite evidence of dynamic malalignment’ and the scores 2 to 4 were categorized as ‘Definite varus present’. Conversion of scores into categorical variables is recommended in order to simplify the ratings (Whatman, Hume & Hing, 2013).

Statistical methods

Statistical analysis was completed using SAS (version 9.1 for Windows; SAS Institute Inc, Cary, NC). Descriptive statistics were calculated for demographics. Weighted kappa values with 95% CIs were used to examine the intratester and intertester reliability of the visual assessment. Cohens weighted kappa values with 95% CIs were used to examine the intratester and intertester reliability of the visual assessment. Cohens weighted kappa is broadly used and is a robust statistic useful for interrater and intrarater reliability testing (McHugh, 2012). Agreement was assessed using the classification for each movement test and for raw data (scores from −4 to 4). Interpretations of Kappa values were based on the guidelines adapted from Landis & Koch (1977): <0.20: Poor agreement; 0.21–0.40: Fair agreement; 0.41–0.60: Moderate agreement; 0.61–0.80: Good agreement; 0.81–1.00: Very good agreement. We aimed at including 60 participants which gave 80% power to detect a kappa-coefficient of at least 0.5, which corresponds to moderate agreement.

Ethical considerations

Ethics approval was given by the Health Research Ethics Committee of the Capital Region of Denmark (De Videnskabsetiske Komitéer for Region Hovedstaden), protocol #H-16045755. Participants written informed consent were obtained prior to the start of the study.

Results

The first sixty individuals with PFP who were included in the parent trial accepted the invitation to participate in this agreement study. Their characteristics are shown in Table 1. The dispersion of the data from the intra- and interrater assessments are presented in a heat map in Fig. 2 (SLS) and Fig. 3 (FL). The weighted kappa values for the classifications and the raw scores are shown in Tables 2 and 3. In summary, the intrarater agreement were statistically significantly different from 0 (p < 0.0001) and ranged from 0.58 to 0.70, i.e., moderate to good agreement, whereas the interrater agreement ranged from 0.22 (p = 0.08) to 0.50 (p < 0.0001), i.e., fair to moderate agreement. Interrater agreement was generally not as good as intrarater agreement (0.7 for SLS intrarater classification scores vs. 0.22 for interrater scores, and 0.58 for FL intrarater classification scores vs. 0.48 for interrater scores). The mean time from baseline to re-evaluation in the intrarater assessment was 29.1 days (SD 14.8). The cross tabulated agreements in the classifications and raw scores are provided in the Supplementary File.

Table 1 Descriptive characteristics of the participants (n = 60).

Characteristics	Mean (SD)	
Age (yrs)	27.2 (6.2)	
Females (n (%))	42 (70%)	
Height (cm)	172.1 (8.6)	
Weight (kg)	66.6 (10.6)	
Body mass index	22.4 (2.8)	
Duration of symptoms (months)	34 (34)	
Anterior Knee Pain Score (0–100 points)	30.20 (5.15)	
Average pain during previous 4 weeks (NRS* 0–10)	3.73 (2.17)	
Notes.

*Numeric rating scale.

Figure 2 Heatmap of agreement matrix showing the dispersion of the intrarater (left) and interrater agreement (right) for the single leg squat.

The brightness of the blue color indicates the number of rating combinations with darker colors representing higher numbers as shown in the individual squares and in the key bar.

Figure 3 Heatmap of agreement matrix showing the dispersion of the intrarater (left) and interrater agreement (right) for the forward lunge.

The brightness of the blue color indicates the number of rating combinations with darker colors representing higher numbers as shown in the individual squares and in the key bar.

Table 2 Intra- and interrater agreement of single leg squat.

	Weighted kappa value	95% CI	p-value	
Intrarater classification	0.70	0.51–0.89	<.0001	
Intrarater raw data	0.65	0.54–0.76	<.0001	
Interrater classification	0.22	−0.03–0.48	0.08	
Interrater raw data	0.32	0.16–0.48	0.05	

Table 3 Intra- and interrater agreement of forward lunge.

	Weighted kappa value	95% CI	p-value	
Intrarater classification	0.58	0.37–0.79	<.0001	
Intrarater raw data	0.65	0.53–0.78	<.0001	
Interrater classification	0.48	0.25–0.7	0.0002	
Interrater raw data	0.50	0.36–0.64	<.0001	

Discussion

Principal findings

The aim of this study was to determine the intra- and interrater agreement between two experienced physiotherapists when visually assessing the dynamic knee alignment during an SLS and an FL in a population of PFP patients. The most important finding was that the visual assessments of dynamic alignment during SLS and FL can be done reliably when the assessment is repeated by the same rater. Moreover, ‘moderate’ levels of agreements are seen when two experienced raters assess the FL, while the interrater agreement is only ‘fair’ when assessing the SLS.

Comparison with previous studies

The intrarater agreements for the SLS in the current study compare to the results of a review on the interrater and intrarater agreement of observation-based assessment of the SLS including studies on both healthy subjects and subjects with lower extremity disorders (Ressman, Grooten & Barr, 2019). In that study, the pooled results of Kappa showed a ‘substantial’ agreement for intrarater agreement (Kappa value 0.68 (95% CI [0.60–0.74]). Moreover, the review found a ‘moderate’ agreement for interrater reliability of the SLS (Kappa value 0.58 (95% CI [0.50–0.65]), which is somewhat higher than in present study. In the present study no efforts were made to synchronize assessors by mutual training sessions or by operationalizing the measurements. Visual assessments were therefore entirely based on the experience of the assessors. This might explain the discrepancy with the systematic review and the relatively low interrater agreements seen.

Clinical implications

The clinical implications of the results of our study are, that visual assessment of frontal plane knee kinematics during a FL can be done reliably by experienced testers, whereas SLS should preferably be re-evaluated by the same tester. Forward lunge is therefore a reliable clinical tool for evaluating knee alignment. Furthermore, it is an inexpensive and easy method to use, making it ideal for the clinical setting. However, the clinical validity, relevance, and prognostic value still need to be established.

Strengths and limitations

This study has some strengths and limitations. Firstly, by using an objective measurement tool (the tablet) we made sure that assessments were based on the same movement, and without verbal or non-verbal interaction between raters. Using the tablet, on the other hand, implies an inherent limitation by not taking the variability of the patients’ performances into account. It was out of the scope of this study to include a clinical re-evaluation of the participants and we thereby accept the exclusion of data on within subject variability. On the other hand, this enables a focused analysis of the agreement within and between raters.

Another strength is, that we have included a relatively large sample of PFP patients, which makes the results transferable to other clinical and experimental settings. It should be noted, though, that assessors were experienced, and our results may only be transferable when assessments are made by equally experienced assessors.

In our assessment of movement quality, we only included the knee excursion during the movement (dynamic knee valgus or varus). Rating dynamic knee alignment per se is not an exhaustive evaluation of movement quality and is merely one aspect of the full range clinical examinations. It is, however, considered a good indicator of movement quality (Sahrmann, Azevedo & Dillen, 2017) with a knee-medial-to-foot position often considered less optimal, indicating poor postural orientation (Ageberg et al., 2010; Ortqvist et al., 2011). A limitation of assessing movement quality without three-dimensional analyses is, that we lack information on movement components like transversal and sagittal plane control of body segments. However, we aimed solely at assessing the reliability of the test assessments and not on the validity of the actual assessments. Furthermore, we did not control for the speed or acceleration of the movement. We intentionally omitted the control in order to comply with our intents of resembling a clinical setup. We acknowledge the limitation of not including the control of speed and acceleration of the execution of the exercises and that speed and acceleration may have influenced the knee excursion.

Various scoring systems have been used to assess dynamic knee joint alignment in the literature (Ressman, Grooten & Barr, 2019). The scoring system used in this study resembles previous used systems (Chmielewski et al., 2007; Whatman, Hing & Hume, 2012). Chmielewski et al., (2007) used the terms “no deviation from neutral alignment”, “small-magnitude or barely observable movement out of a neutral position”, “moderate-magnitude or marked movement out of a neutral position, and “excessive or severe magnitude of movement out of a neutral position”, to assess the degree of knee excursion during a unilateral squat and lateral step-down task. We made the scoring two-tailed (varus and valgus) to be more specific in the direction of the knee in the movement. The use of different scoring systems in the literature, makes it difficult to compare and pool the results of agreement studies. Future studies should consider standardizing the scoring for the benefit of research in reliability and agreement of knee kinematics.

No varus knees (>1 on the scoring scale) were found in the population. This means that when scores were converted into classifications there were only two viable options so it is difficult to tell if the results would be similar if individuals with varus were included in the analysis. The narrow range of scores is probably linked to the population, indicating that PFP patients are more prone to a valgus knee alignment.

Comparing real time scoring to retrospective scoring may have impacted our results. We chose the real time visual assessment in order to resemble a clinical setup and accepted the potential bias of re-evaluating the movement on a screen. We argue, however, that since the assessment was only in the frontal plane, the risk of bias in the subsequent re-rating on a screen was small. Further, the re-assessment of the video recordings did not include slow motion, which makes the validity of the results representative of clinical practice.

Conclusions

Visual assessments of dynamic knee joint alignment during a FL and a SLS performed by patients with PFP can be done reliably when an experienced rater repeats the assessment. Two experienced physiotherapists agree moderately when assessing dynamic alignment during FL, but only ‘fair’ when assessing dynamic alignment during SLS. The agreement within raters is better than between raters, which suggests that assessments should preferably be performed by the same tester in research and in a clinical setting, e.g., to evaluate any treatment effect. We suggest the FL as a reliable clinical tool for evaluating knee alignment in the clinical setting, since it shows acceptable intra- and interrater agreement, and it is an inexpensive and easy test to use.

Supplemental Information

Supplemental Information 1 Cross tabulated agreements in the classifications and raw scores

Click here for additional data file.

Supplemental Information 2 Raw data for ratings of single leg squat and forward lunge

Click here for additional data file.

Additional Information and Declarations

Competing Interests

Author Contributions

Human Ethics

Data Availability

The authors declare there are no competing interests.

Rudi Hansen conceived and designed the experiments, performed the experiments, analyzed the data, prepared figures and/or tables, authored or reviewed drafts of the paper, and approved the final draft.

Mathilde Lundgaard-Nielsen performed the experiments, analyzed the data, authored or reviewed drafts of the paper, and approved the final draft.

Marius Henriksen conceived and designed the experiments, analyzed the data, prepared figures and/or tables, authored or reviewed drafts of the paper, and approved the final draft.

The following information was supplied relating to ethical approvals (i.e., approving body and any reference numbers):

Ethics approval was given by the Health Research Ethics Committee of the Capital Region of Denmark, protocol #H-16045755.

The following information was supplied regarding data availability:

The cross tabulated agreements in the classifications and raw scores are available in the Supplementary File.

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
