# Peer review of "Visual assessment of dynamic knee joint alignment in patients with patellofemoral pain: an agreement study"

_PeerJ, doi:10.7717/peerj.12203_

## Round 0.1 · original submission · Major Revisions

The main concerns of the research are the large number of days between sessions for the assessment of intra-rater reliability ("The mean time from baseline to re-evaluation in the intrarater assessment was 29.1 days (SD 14.8).") How can the authors justify this situation? The methodological recommendations to assess intra-rater reliability is not to exceed 7 to 10 days between sessions.

On the other hand, the methodology does not present a control/comparison with a gold standard that allows to specify whether indeed the participants presented a kinematic alteration. Furthermore, the authors did not control the speed or acceleration of execution of the exercises. These could be factors in the alignment of the lower limb.

Reviewer 1 ·

Basic reporting

Table 1. Describe the training experience of participants.
Line 132. Indicate the frames per second of the video records.
A hot map figure (with individual data) can be helpful to understanding the dispersion of your data.

Experimental design

My principal concern about the methodology used in the present study is that the dynamic knee joint alignment scoring was not contrasted with a gold standard (e.g., kinematic). The agreement to the visual evaluation of dynamic knee joint alignment was made based on an assessment by two experts, limiting the extrapolation of results (e.g., professionals < 5 yr experience).
Line 182. Can you indicate if this scoring used is validated, or has it been used in a previous study?.
Line 146 and 154. Did the author controlled/registered de velocity of the tests?. Did the participant practiced the tests before to applied the measures?.
Line. 184. Explain how the sample size was chosen. State the determined number of raters, subjects/objects, and replicate observations.
Line 246. The limitations must be expanded.

Validity of the findings

'no comment'

Additional comments

This paper is well written and is an interesting topic worthy of further investigation. Importantly, however, the results must be more clearly presented. More information about the protocol is needed.

Reviewer 2 ·

Basic reporting

no comment

Experimental design

Page 9, line 180, missing a quotation mark after the word malalignment

There are also a few things that need to be clarified about the methology:

The participants performed the exercises without practicing it, however, there may have been some among the participants who were familiar with the gesture, which could have been a determining factor in their quality of execution due to prior motor learning. Were the study subjects sedentary or physically active?

Were the exercises performed at a speed selected by the participant? How was the influence of acceleration on the knee excursion controlled? This factor could influence the knee valgus.

What criteria did the authors use to categorize as severe, moderate, mild and doubtful varus / valgus? Does this represent an adaptation of the system used by Nae et al (2017)? It is necessary to clarify this to allow the reproducibility of your system, considering that subjects with doubtful varus / valgus were finally categorized as "No definite evidence ...". The authors could give an example or brief description of this categorization, otherwise it is too dependent on the judgment of the evaluator, independent of their clinical experience.

Validity of the findings

Page 11, line 242. The authors state "Forward lunge is therefore a promising clinical tool for evaluating knee alignment". Since this study deals with intra- and inter-rater reliability, this statement cannot be attributed to the results. The authors did not measure the sensitivity or specificity of the test. I understand that you meant that forward lunge is a reliable test to be used in clinic enviroment, as you say later.

An element to note is that the video analysis carried out did not include slow motion. This makes the validity results more representative of clinical practice than other studies.

Additional comments

It is a good article, clear and consistent between its objective and methodology. The authors' interest in bringing research closer to daily clinical aspects is appreciated.

There are methodological appraisals that must be reviewed, mainly considering the reproducibility of the classification system they propose.

---

## Round 0.2 · accepted · Accept

The authors fully addressed the comments and suggestions of both reviewers. Especially the methodological aspects of research and the recognition of the inherent limitations of research design.

Reviewer 1 ·

Basic reporting

I believe the authors have satisfactorily addressed my comments and I recommend that the article be accepted for publication.

Experimental design

I have no comments

Validity of the findings

I have no comments

Additional comments

I have no comments.

Reviewer 2 ·

Basic reporting

no comment

Experimental design

no comment

Validity of the findings

no comment

Additional comments

The authors made any necessary modifications in consideration of the comments of the previous review. The article is better understood from a methodological perspective